# MicroRNA Interrelated Epithelial Mesenchymal Transition (EMT) in Glioblastoma

**DOI:** 10.3390/genes13020244

**Published:** 2022-01-27

**Authors:** Botle Precious Setlai, Rodney Hull, Rui Manuel Reis, Cyril Agbor, Melvin Anyasi Ambele, Thanyani Victor Mulaudzi, Zodwa Dlamini

**Affiliations:** 1Department of Surgery, Level 7, Bridge E, Steve Biko Academic Hospital, Faculty of Health Sciences, University of Pretoria, Private Bag X323, Arcadia 0007, South Africa; cyrilagbor0772@gmail.com (C.A.); thanyani.mulaudzi@up.ac.za (T.V.M.); 2SAMRC Precision Oncology Research Unit (PORU), Precision Oncology and Cancer Prevention (POCP), Pan African Cancer Research Institute (PACRI), University of Pretoria, Hatfield 0028, South Africa; rodneyhull@gmail.com (R.H.); ruireis.hcb@gmail.com (R.M.R.); 3Molecular Oncology Research Center, Barretos Cancer Hospital, Antenor Duarte Villela, 1331, Barretos 14784-400, SP, Brazil; 4Life and Health Sciences Research Institute (ICVS), School of Medicine, University of Minho, 4710-057 Braga, Portugal; 5Department of Oral Pathology and Oral Biology, School of Dentistry, Faculty of Health Sciences, University of Pretoria, P.O. Box 1266, Pretoria 0001, South Africa; melvin.ambele@up.ac.za; 6Institute for Cellular and Molecular Medicine, SAMRC Extramural Unit for Stem Cell Research and Therapy, Department of Immunology, Faculty of Health Sciences, University of Pretoria, Pretoria 0001, South Africa

**Keywords:** microRNA, EMT, angiogenesis, drug resistance

## Abstract

MicroRNAs (miRNA) are small non-coding RNAs that are 20–23 nucleotides in length, functioning as regulators of oncogenes or tumor suppressor genes. They are molecular modulators that regulate gene expression by suppressing gene translation through gene silencing/degradation, or by promoting translation of messenger RNA (mRNA) into proteins. Circulating miRNAs have attracted attention as possible prognostic markers of cancer, which could aid in the early detection of the disease. Epithelial to mesenchymal transition (EMT) has been implicated in tumorigenic processes, primarily by promoting tumor invasiveness and metastatic activity; this is a process that could be manipulated to halt or prevent brain metastasis. Studies show that miRNAs influence the function of EMT in glioblastomas. Thus, miRNA-related EMT can be exploited as a potential therapeutic target in glioblastomas. This review points out the interrelation between miRNA and EMT signatures, and how they can be used as reliable molecular signatures for diagnostic purposes or targeted therapy in glioblastomas.

## 1. Introduction

Gliomas are the most common primary carcinomas of the central nervous system (CNS), arising from structural supporting brain cells known as glial cells. Glial cells have been shown to play an important role in the development and function of the CNS. Patients with gliomas have an average survival time of 12–18 months and a 5-year patient survival rate of 5% [1]. Glioblastomas are the most invasive type of gliomas, with a poor prognosis and few treatment options. For most of these patients, the treatment is mostly palliative [2]. The incidence of the disease is higher in men in their mid-50s than in women. Glioblastomas occasionally arise from genetic syndromes or from exposure to ionizing radiation [3]. The regulatory mechanisms of miRNAs in glioblastoma processes are still not fully elucidated. Several miRNA signatures have been associated with either tumor suppression or promotion. Glioblastomas express several miRNAs, which have been shown to prevent progression of the disease whilst facilitating apoptosis. An occurrence was observed following treatment with chemotherapeutic agents. In this case, some of the miRNAs acted as apoptotic genes, suppressing the proliferation and survival of glioblastomas or silencers of anti-apoptotic genes, thus favoring tumor growth. This potentiates the use of miRNA signatures as diagnostic biomarkers for glioblastoma, primarily with the use of liquid biopsies, which are easily obtainable [4]. miRNAs have been shown to regulate epithelial to mesenchymal transition (EMT) [5], a process whereby epithelial cells undergo biological processes that allow them to take on the phenotype of mesenchymal stem cells and adopt their enhanced migratory ability, invasive capacity and increased ability to resist apoptosis [6]. EMT is closely associated with malignant progression and clinical outcome in gliomas [7]. Manipulation of miRNA-EMT regulatory mechanisms for treatment purposes includes targeting EMT transcription factors and structural components of the epithelial cells in an attempt to reverse the EMT processes, as well as targeting the miRNA signaling pathways involved in these processes [8]. This review addresses miRNAs that play a role in regulating EMT signaling pathways and how they can be targeted as biomarkers or potential targets for the treatment of glioblastomas. As glioblastomas are a type of gliomas, the two might be discussed interchangeably in cases where the effect is expected to be the same.

## 2. Epithelial Mesenchymal Transition in Glioblastoma

During EMT, epithelial cells develop a mesenchymal phenotype and properties including migration. Cancer cells acquire these abilities and migrate to distant sites away from the primary tumor to establish metastatic foci. The process of EMT and the resultant migratory abilities are also associated with drug resistance. A number of processes contributing to glioma EMT have been described. The induction of these processes in glioblastomas differ from those observed in epithelial cancers because of the absence of the basement membrane [9,10]. The basement membrane is a specialized extracellular matrix that forms the base for endothelial and epithelial cells. It plays a pivotal role in maintaining the structural integrity, exchanging biochemical signals, and regulating cellular function [11,12]. Typically, for tumor cells to migrate to other parts of the body, they need to detach from the basement membrane. In the case of gliomas, glial cells are the most abundant cells in the brain, and provide protective and structural support [13]. A type of glial cell, referred to as astrocyte, becomes activated and surrounds the tumor stroma, thus keeping cancer cells and the cluster intact. These cells can undergo EMT processes induced by cancer cells and EMT-associated transcription factors [14] (Figure 1). Astrocytes have also been shown to develop aberrant signaling pathways that are capable of inducing cancer cell migration and metastasis. They can also protect the brain from cancer by secreting molecules that keep the blood–brain barrier intact and prevent invasion by cancer cells [15].

## 3. Drivers of EMT in Glioblastoma

### 3.1. Extracellular Vesicles (EVs)

Extracellular vesicles secreted by tumors facilitate the intercellular signaling and communication within the components of the tumor microenvironment and the tumor stroma [16]. EVs have been shown to carry a number of biological molecules, including miRNAs [17], hence their involvement in the transfer of mutations. Furthermore, EVs are involved in cell proliferation, migration and homing, reprogramming energy metabolism, angiogenesis and drug resistance [16]. Glioblastomas have been classified into proneural (PN), mesenchymal (MES) and the classical tumor-intrinsic transcriptional subtypes implicated in aiding drug resistance [18]. MES are known to play an important role in EMT induction [19]. Thus, EVs originating from MES drive PN glioblastoma progression and, ultimately, drug resistance through an NF-κB signal transducer and STAT3 signaling [20]. EVs have also been found to be major players of drug resistance in glioblastomas by transporting anti-cancer agents out of the tumor microenvironment [21], highlighting the importance of targeting EVs in the endeavor of developing novel therapeutic strategies for this disease. miRNAs using EVs as their nano-vehicles in glioblastomas have been thoroughly reviewed elsewhere [22].

### 3.2. Transforming Growth Factor β Signaling Pathway

Transforming growth factor-β (TGF-β) is a family of multi-functional growth factors involved in cell proliferation, differentiation and apoptosis. During cancer development, TGF-β suppresses cancer growth via cell cycle arrest and apoptosis, but supports cancer progression at a later stage of the disease by promoting invasion, increasing migration capabilities and drug resistance [23]. The TGF-β complex consists of surface receptor type I (TβRI) and type II (TβRII) transmembrane serine/threonine kinases. Signaling is initiated by the binding of TGF-β to TβRII and TβRI receptors on the cell surface, resulting in a heterocomplex. Phosphorylation by TβRII activates TβRI, resulting in the recruitment and phosphorylation of Smad proteins 2 and 3, collectively known as R-Smads. Translocation to the nucleus requires the binding of R-Smads to Smad 4. The resultant R-Smads/Smad 4 complex will then induce the transcription of target proteins [24] (Figure 2). Aberrant TGF-β signaling pathways contribute to cancer development [25], making TGF-β an important driver of EMT in cancers.

The mesenchymal glioblastoma has the most severe clinical outcome compared to the other subtypes. Joseph et al. reported that the activation of Smad 2 and ZEB1 promotes the progression of glioblastomas into the most aggressive, drug-resistant, mesenchymal subtype [26]. ZEB1 is a transcription factor known to facilitate mesenchymal transition in epithelial cancers, resulting in invasion and metastasis [27]. TGF-β was shown to transcriptionally upregulate PDK1, which, in turn, upregulates c-Jun. The resultant PDK1/c-Jun pathway facilitates the EMT process, which then promotes cancer cell proliferation and invasion in glioblastomas [28]. Previous studies showed that E2 induces cell proliferation and the expression of genes involved in mitochondrial metabolism in glioblastomas [29]. The treatment of glioblastoma cells with E2 resulted in decreased expression of the TGF-β intracellular signaling proteins R-Smads [30]. In another study, TGF-β1 induced activation of the Wnt/β-catenin pathway during the EMT processes. This resulted in increased expression of EMT factors, such as vimentin, N-cadherin, Β-catenin and cyclin-D1, in human glioma cells. However, the treatment of glioma cells with astragaloside IV neutralized this effect [31]. These studies suggest that both E2 and astragaloside IV could serve as the potential treatment of choice for glioblastoma. 

### 3.3. Autophagy

Autophagy is a homeostatic process that clears out non-essential cellular components through lysosome-mediated intracellular degradation activity [32]. Cancer cells use autophagy as a gateway to evade oxygen/nutrient deprivation and pharmacotherapy. Autophagy also serves as a source of the energy they require for rapid proliferation and survival. Gugnoni et al. reviewed the interconnection between autophagy and EMT in cancer. Here, the authors point out the intricate relationship between the two processes, whereby autophagy has a dual effect on EMT. The activation of autophagy can suppress the EMT processes at the beginning of metastasis, but can turn around and enhance these activities during the EMT processes [33]. A number of signaling pathways, as well as biological factors, have been implicated in autophagy–EMT interplay. Sun et al. noted a positive correlation between activated autophagy and the TGF-β2/RSmads signaling pathway, which resulted in enhanced EMT processes [34]. The same effect was observed in glioma cells, with additional inhibition of c-Jun NH2-terminal kinase, which negatively correlated with TGF-β2-activated autophagy, resulting in reduced EMT processes. The study indicated that autophagy is essential for TGF-β-induced glioma invasion [35].

The NADPH oxidases (NOXs) of genes are the main suppliers of reactive oxygen species (ROS). The transcription NOX4 subfamily mRNA has been implicated in the proliferation and survival of glioblastoma cells. Together with TGF-β, NOX4 enhanced glioblastoma growth. Patients with higher levels of TGF-β and NOX4 had a poor prognosis, suggesting that the two molecules could serve as useful biomarkers of the disease [36]. Wnt signaling pathways are involved in embryonic developmental processes and stem cell proliferation in metazoan animals [37]. The pathway has a dual role in cancer, being composed of a myriad of oncogenes and tumor suppressors, with genes coding for the components of the anaphase-promoting complex (APC) being the most frequently mutated [38]. Autophagy is divided into microautophagy, macroautophagy and chaperone-mediated autophagy (CMA). During microautophagy, the lysosomal membrane bulges and encapsulates the targeted molecule by invagination. In macroautophagy, the lysosomal membrane surrounds the targeted molecule and draws it into the lysosome. Conversely, the protein substrates of CMA are recognized and targeted by KFERQ pentapeptide motifs, which are then escorted across the lysosomal membrane [39,40]. Coelho et al. reviewed the relationship between EMT and autophagy in glioblastoma. The authors indicted that aberrant Wnt signaling promotes development, progression and invasiveness in glioblastomas. Wnt signaling further promotes the EMT process, whilst suppressing autophagy. The TGF-β2/RSmads signaling pathway induced autophagy in gliomas, and has been reported to take place through AEG-1/MTDH activation [41], identified as a prominent oncogene [42]. Autophagy is essential in TGF-β-induced glioblastoma, whilst TGF-β is one of the most noticeable drivers of EMT, emphasizing the importance of the autophagy/TGF-β/EMT pathway as a therapeutic target for glioblastoma.

### 3.4. MiRNAs

The expression signatures of a number of miRNAs have been identified as either instigators or inhibitors of EMT in glioblastomas. Zhang et al. assessed the clinical importance of EMT in glioma using independent glioma datasets, GSE16011, Rembrandt and The Cancer Genome Atlas (TCGA) program. The study found 19 miRNA expression signatures that are positively correlated with EMT-driven gliomas, with the most prominent being miR-223 [7]. Blocking the miR-223/PAX6 pathway improved sensitivity to chemotherapeutic treatment with temozolomide in glioblastomas, indicating the role of miR-223 in drug resistance [43]. The PI3K/Akt signaling pathway is associated with EMT-related disease severity in glioblastomas. This pathway has been shown to act in collaboration with other EMT-related signaling pathways, including mTOR [44], CXCR4 signaling [45] and LASP1 [46]. Huang et al. noted that the miR-223/PAX6 axis promoted disease progression and drug resistance by the activation of the PI3K/Akt signaling pathway, suggesting the possible use of this pathway as a novel potential therapeutic intervention for glioblastoma [43].

Figure 3 shows some of the miRNAs that play a role in the proliferation of glioblastoma. The miR-223 gene is transcribed into two separate miRNAs, using the plus and minus strands, namely, miR-223-5p and miR-223-3p, which have been implicated in glioblastomas. Contrary to the previous studies, the treatment of glioblastoma cells with the miR-223-3p analogue suppressed the proliferation and migration of cancer cells, indicating the potential use of miR-223-3p for targeted therapy [47]. Zhang et al. found that of the 18 miRNAs that negatively correlated with glioblastoma, miR-95 was the most prominent [7]. The miRNA profile of stem cells isolated from primary human glioblastoma tissue was compared to autologous differentiated tumor cells, and it was found that miR-95 and miR-21 displayed the most aberrant features. In this study, miR-95 downregulation was related to improved clinical outcome in the neural subtype, an effect shared with the upregulation of miR-21 [48]. The upregulation of miR-381 was also found to effectively inhibit EMT and metastasis by downregulating the transcription factor lymphoid enhancer-binding factor 1 (LEF-1) in glioblastoma. This transcription factor is involved in the Wnt signaling pathway [49].

Many of the miRNAs identified as playing a role in promoting glioblastoma are involved in regulating the levels and activities of multiple cell surface receptors. Epidermal growth factor receptor (EGFR) is known to promote glioblastoma proliferation. The expression of EGFR is regulated by multiple miRNAs, including miR-7-5p [50], miR-491-5p [51], miR-218 [52] and miR-128-5p [53]. miR-218 is also able to promote glioblastoma by targeting the components of receptor tyrosine kinase (RTK) signaling pathways; this results in increased hypoxia-inducible factor 2α (HIF2α) activity [52]. miR-128-3p promotes glioblastoma by inducing the expression of the platelet-derived growth factor α receptor [53]. This is one of the functions shared with miR-34a-5p in promoting glioblastoma development and progression. miR-34a-5p increases the expression of the tyrosine kinase receptor and the hepatocyte growth factor receptor [54]. Other miRNAs promote glioblastoma proliferation by affecting the activity or expression of molecules in the RAS signaling pathway, or through the inhibition of tumor suppressors. Neurofibromin (NF1) is a cytosolic protein that promotes the GTPase activity of RAS, increasing intracellular activity. The activity of NF1 is downregulated by miR-9-5p [55]. The expression of separate components of the RAS pathway is also increased through the action of multiple miRNAs. These include let5a-5p, which targets KRAS [56], miR-143-3p, which targets NRAS [57], and miR-123-3p, which targets HRAS, NRAS and KRAS [58].

The analysis of miRNA expression signatures, with an intent to use them as independent predictors of clinical outcome in glioblastoma, identified ten miRNAs (miR-17-5p, miR-20a, miR-31, miR-106a, miR-146b, miR-148b, miR-193a, miR-200b, miR-221 and miR-222). The study showed that patients with high risk had shorter survival times than patients who were in the low-risk group. The risk score was constructed according to the ten miRNA expression signatures [59]. Thus, from their potential use as molecular biomarkers, targets for therapy or prognostic markers, the role of miRNA in glioblastoma is prominent, and more studies are needed to stratify the application of miRNA in the continuous fight against one of the deadliest cancers of the central nervous system.

## 4. miRNA-EMT-Related Cancer Cell Invasion and Metastasis

The link between miRNA expression signatures, EMT and, ultimately, cancer metastasis has long been established [60]. Several miRNA signatures are involved in EMT-associated mechanisms of invasion and metastasis in glioblastomas. Studies on miR-125a-5p found its expression to suppress the mesenchymal capabilities of glioblastoma cells, thus inhibiting EMT [61,62]. Recently, Nan et al. showed that miR-451 expression is able to suppress EMT and metastasis by blocking the PI3K/Akt/Snail signaling pathway through the activation of calcium-binding protein 39 (CAB39) in gliomas [63]. The expression of miR-451 was associated with the inhibition or reversal of the EMT processes in cancers [64]. Contrary to this finding, miR-451 expression plays different roles in gliomas, as it was associated with the proliferation of cancer cells by suppressing the CAB39/AMPK/mTOR pathway, and increased metastatic potential, which takes place via the activation of the Rac1/cofilin pathway [65]. The increased expression of miR-200b-3p prompted E-cadherin (essential for maintaining the structural integrity of epithelial cells) by the deactivation of extracellular signal-regulated kinase 5 (ERK5), resulting in reduced glioma cell proliferation and mesenchymal capabilities [41]. mR-424 showed promising results as a potential prognostic molecular marker and therapeutic target by blocking EMT processes and the resultant metastatic capabilities by targeting the kinesin-like protein KIF23 in human glioma [66]. The same effect was observed with miR-378, which inhibited EMT by targeting cis-aconitate decarboxylase (IRG1) in gliomas [67], miR-139-5p by targeting the notch1 oncogene in gliomas [68], miR-181a by targeting ZBTB33 expression in glioma cells [69], miR-623 by targeting TRIM-44 [70], miR-940 by targeting ZEB2 [71], and miR-7, which targeted T-Box 2 in glioblastoma [72]. In addition to these miRNAs, miR-182-targeted MTSS1 enhanced TGF-β1-related EMT in gliomas [73], and miR-504 inhibited EMT in glioblastomas by targeting the Wnt receptor FZD7/β-catenin pathway. The phosphatidylinositol 3,4,5-trisphosphate 3-phosphatase and dual-specificity protein phosphatase (PTEN) tumor suppressor inhibits PI3K by dephosphorylating it. The expression of PTEN is inhibited through the action of three separate miRNAs. These are miR-17-5p [74], miR-23a-3p [75] and miR-26a-5p [76]. The p53 tumor suppressor is, itself, downregulated by miR-10b-5p. This miRNA also downregulates the tumor suppressor and cell cycle regulator p16 [77]. At the same time, the negative regulator of p53, MDM2, is upregulated in glioblastoma by miRNAs such as miR-32-5p, miR-25-3p and miR-17-3p [78]. The RB1 tumor suppressor is downregulated by the miRNA miR-28-5p. At the same time, the expression and activity of oncogenes that promote cell cycle progression are downregulated by various miRNAs in glioblastoma; these include miR-124-3p suppressing cyclin-dependent kinase (CKD) 4 [79], and miR-491-5p, miR-491-3p and miR-138-5p [80] suppressing CKD6. Finally, the proliferative activity of cyclin D is downregulated by miR-195-5p [81]. A decreased expression ratio of the miRNA miR-504/FZD7 was shown to be a potential molecular marker for identifying the mesenchymal subtype in glioblastoma [82]. The actions of the miRNAs discussed above are summarized in Figure 4.

## 5. miRNA-EMT-Related Angiogenesis

Glioblastomas are solid tumors that require blood supply for the delivery of oxygen, energy, and the nutrients needed for them to survive and thrive. To achieve this, tumor cells develop mechanisms to initiate the development of new blood vessels, a process known as neovascularization, which is covered in greater detail elsewhere [83]. Angiogenesis is one of the mechanisms that tumor cells use in this regard. It is a process where new blood vessel growth is stimulated by the secretion of proangiogenic factors, such as vascular endothelial growth factor (VEGF), in response to relative hypoxic conditions in areas within the tumor where there is relatively little blood supply. In the absence of angiogenesis, cancer cells become necrotic or apoptotic, hence angiogenesis is pivotal in cancer cell survival and progression. This is one of the main reasons why antiangiogenic therapy, which targets pro-angiogenic factors and their signaling pathways, was developed. However, the efficacy of this treatment is still inadequate, as tumor cells are able to find other ways to promote angiogenesis and resistance to anti-angiogenic drugs [84]. More and more studies are dedicated to exploring multiple pathways to inhibit the mechanisms that tumor cells use to induce angiogenesis. This includes, but is not limited to, EMT-induced angiogenesis.

MicroRNAs regulating angiogenesis in cancers have been dubbed angiomiRNAs [85,86]. Several angiomiRNAs have been identified in glioblastomas [87], with some originating from glioblastoma-derived EVs [88]. However, studies indicating the relationship between angiomiRNAs and the EMT process in glioblastomas are limited. Dai et al. found that upregulated miR-24 expression induced the expression of the angiogenic markers VEGF and, TGF-β, and matrix metalloproteinases (MMP)-2 and -9, resulting in increased glioblastoma cell proliferation and development [89] (Figure 5). Increased expression of miR-16 in glioblastoma inhibited EMT processes by targeting polycomb complex protein BMI-1, and reducing the expression levels of the angiogenic markers VEGF-A and VEGF-C [90]. miR-576-3p was shown to inhibit EMT and the angiogenic properties of hypoxia-treated glioma cells by targeting HIF-1α [91] (Figure 5). Vosgha et al. also showed this relationship in anaplastic thyroid carcinoma. The study noted that upregulation of miR-205 significantly inhibited angiogenesis and EMT by targeting VEGF and ZEB1, respectively. miR-124-3p downregulates invasion and migration in glioblastoma by targeting β1 integrin [92]. MMPs are required to degrade the ECM, allowing for cell migration and invasion. They are targeted by miR-152-3p, miR-491-5p and miR-211-5p [93]. Reversion-inducing cysteine-rich protein with Kazak motifs (RECK) is a membrane-bound receptor that inhibits metalloproteinase activity, as well as the metalloproteinase inhibitor TIMP3, which are both inhibited by miR-21-5p [94,95]. Disintegrin and metalloproteinase domain-containing protein 17 (ADAM17) is targeted by miR-145. This protein increases invasion and metastasis by cleaving and activating TNFα [96]. The importance of ADAM17 in angiogenesis was demonstrated in a study done by Caolo et al. [97]. The von Hippel–Lindau (VHL) disease tumor suppressor inhibits HIF-1α (via proteosomal degradation), thereby decreasing VEGF expression. VHL is targeted by both miR-21-5p [98] and miR-23b-3p [99]. More studies focusing on angiomiRNAs related to EMT could pinpoint miRNAs that can be targeted to halt and/or prevent the severity of the disease, thus increasing overall survival. Table 1 indicates some of the angiomiRNAs that also play a role in EMT that could further be investigated for management of the disease.

### AngiomiRNA-EMT-Induced Drug Resistance

The potential of Raddeanin A (RA) as treatment in cancers [105,106], including glioblastomas [107] has been promising thus far. One of the mechanisms employed by RA is the inhibition of EMT and angiogenic processes by the deactivation of β-catenin and EMT pathways, which then targets specific molecules (N-cadherin, vimentin and Snail). This process results in reduced glioblastoma cell proliferation, invasion and metastasis [52]. The levels of most miRNAs are typically reduced in gliomas, serving as a major stepping stone for drug resistance. The sensitivity to chemotherapeutic treatment with temozolomide correlated with the overexpression of miR-218 in the mesenchymal subtype. Downregulated miR-218 activates receptor tyrosine kinase signaling pathways, which, in turn, activates HIF2α, resulting in improved cancer cell survival and enhanced angiogenesis [108]. EphrinB2 is a subfamily of receptor protein tyrosine kinases implicated in tumorigenesis. Its downregulation in gliomas is controlled by HIF1α via the activation of ZEB2, resulting in cancer cell invasion and anti-angiogenic resistance. Thus, the study suggests that a combinatorial therapeutic strategy with anti-angiogenic treatment aimed at inhibiting hypoxia signaling pathways could improve clinical outcomes. Researchers also point out the significance of the function of ZEB2 as an attractive therapeutic strategy to prevent EMT activity and control drug resistance [109].

## 6. Conclusions

The use of miRNAs for diagnostic purposes is attractive, as miRNAs are easily attainable through liquid biopsy. The levels of a number of miRNAs are decreased in glioblastomas. These miRNAs are involved in tumor progression, as proven in studies where the overexpression of these miRNAs resulted in the suppression of cancer progression and enhanced apoptosis. Specific miRNAs have great potential to be used as biochemical markers and therapeutic targets in glioblastomas. Several factors are involved in EMT processes, which are considered to be the major contributor to cell proliferation and metastasis in glioblastomas. There is, thus, a greater need to block EMT processes to prevent cancer progression and its spread to distant areas of the brain. Drugs that are aimed at preventing other drivers of EMT, such as angiogenesis, have not been successful in the prevention of cancer progression thus far. Cancer cells develop mechanisms to resist such therapeutic agents. It is, thus, important that more efforts are dedicated to developing combinatorial molecular therapeutic strategies that target miRNAs as drivers of EMT, particularly in combination with other signaling factors, such as TGF-β, and/or angiogenic factors, such as VEGF/HIF and factors involved in autophagy.

## Figures and Tables

**Figure 1 genes-13-00244-f001:**
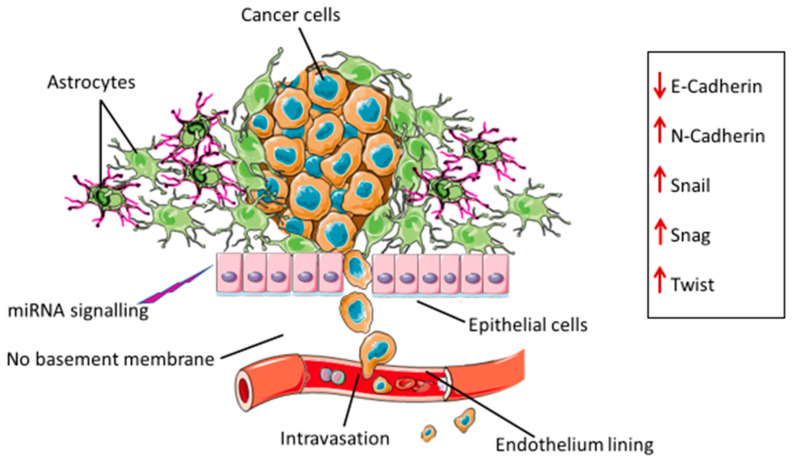
Stimulus from miRNA signaling molecules, EMT transcription factors (N-cadherin, Snail, Snag and Twist) and reduced levels of E-cadherin (needed for maintaining the cell-to-cell adhesion of the epithelial cells) induce the EMT processes. Epithelial cells and astrocytes, which form part of the tumor stroma cells, will then lose their adhesion capacity and acquire a mesenchymal phenotype, leading to cancer cells migrating into and invading the surrounding tissue. Astrocytes can be activated by the tumor. These reactive astrocytes become cancerous and serve as a supporting structure for the cancer cells. At this point, astrocytes can activate the aberrant signaling pathways involved in the induction of EMT, resulting in metastasis.

**Figure 2 genes-13-00244-f002:**
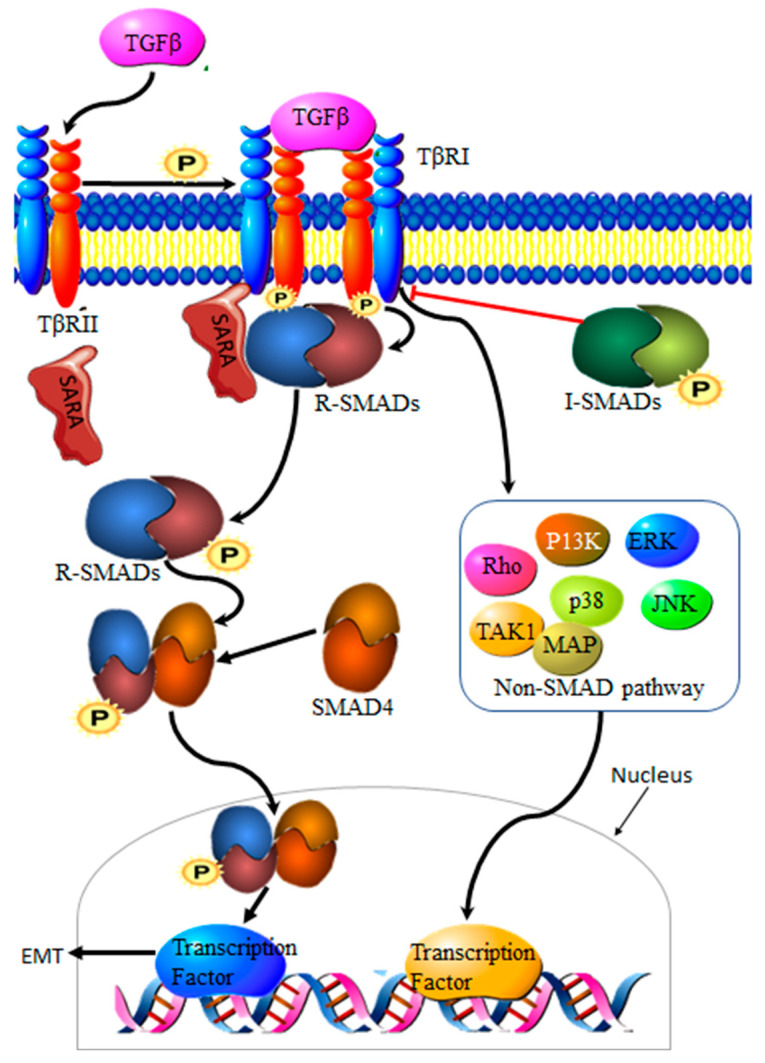
Transforming growth factor-β (TGF-β) signaling pathways; the signaling pathway is initiated by the binding of TGF-β to TβRII and, in the process, recruiting TβRI, forming a heterodimer complex. TβRII will activate TβRI by phosphorylation, resulting in recruitment of R-Smads, which will be anchored onto the complex by SARA. Phosphorylation of R-Smad or receptor-activated Smads will take place. Phosphorylated R-Smads will bind to Smad4, creating a complex that then signals translocation into the nucleus, leading to transcription of target proteins. TGF-β signaling can also result in the activation of non-Smad signaling, which also leads to target gene transcription, while I-Smads act as an inhibitor of the TGF-β signaling pathway.

**Figure 3 genes-13-00244-f003:**
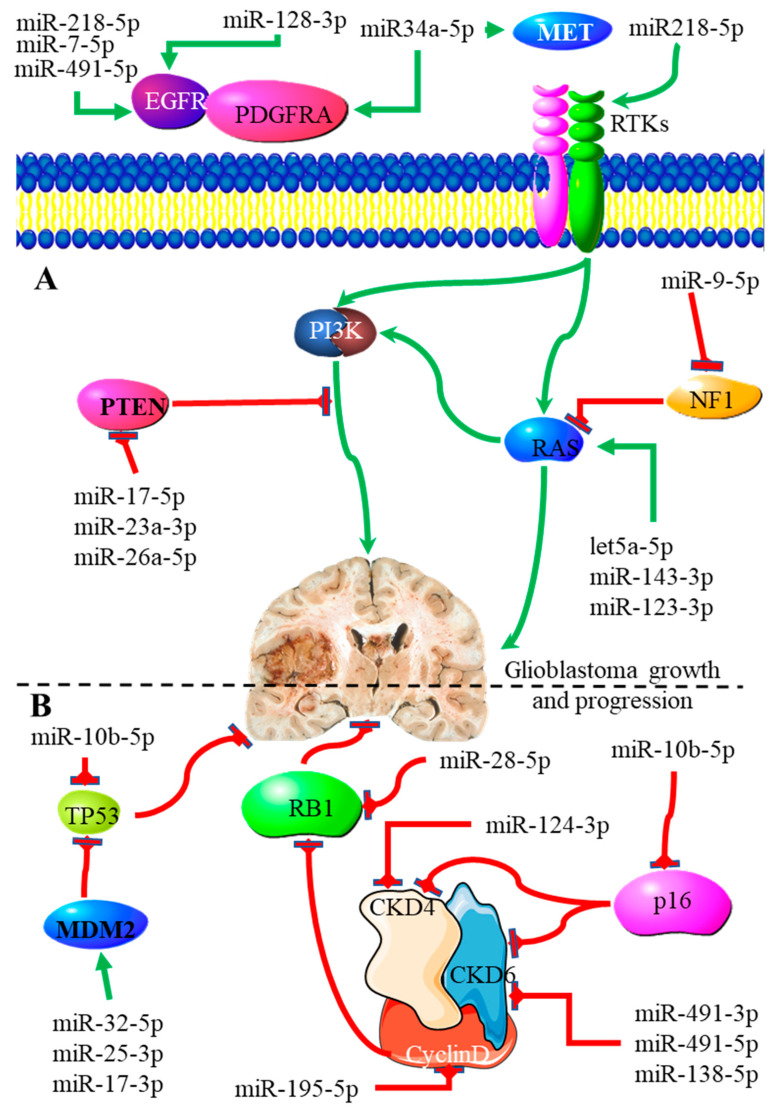
The role played by various miRNAs in glioblastoma (**A**) proliferation and (**B**) evading growth suppression. (**A**) The expression of various ligands that bind to receptor tyrosine kinases is promoted by specific miRNAs. The PTEN signaling pathway, which inhibits proliferation, is downregulated by three specific miRNAs (miR17-5p, miR-23a-3p and miR-26a-5p), while the RAS pathway promoting proliferation is upregulated by miR-143-3p, miR-123-3p and let5a-5p. (**B**) The regulation and signaling of cyclin D are controlled through numerous miRNAs. The ability of p53 to downregulate growth suppression is downregulated by miR10p-5p and MDM2, whose expression is upregulated by various miRNAs.

**Figure 4 genes-13-00244-f004:**
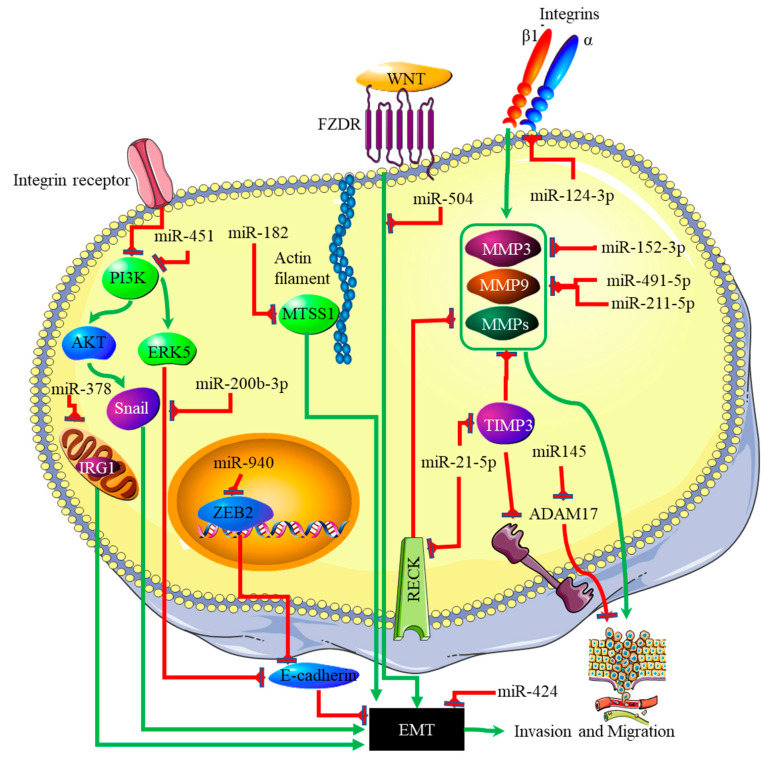
miRNA involved in the invasion and metastasis pathways in glioblastoma; miR-451 blocks the PI3K/Akt/Snail signaling pathway to suppress EMT and metastasis. The expression of E-cadherin maintains the structural integrity of epithelial cells. E-cadherin is repressed by ERK5, while the ERK5 pathway is inhibited by miR-200b-3p, reducing EMT and glioma cell proliferation. The Wnt pathway, which promotes EMT and invasion, can be blocked by miR-504, which blocks the Wnt receptor FZD7. The expression of various matrix metalloproteinases and the metalloproteinase inhibitor 3 (TIM3) can also be controlled by miRNAs.

**Figure 5 genes-13-00244-f005:**
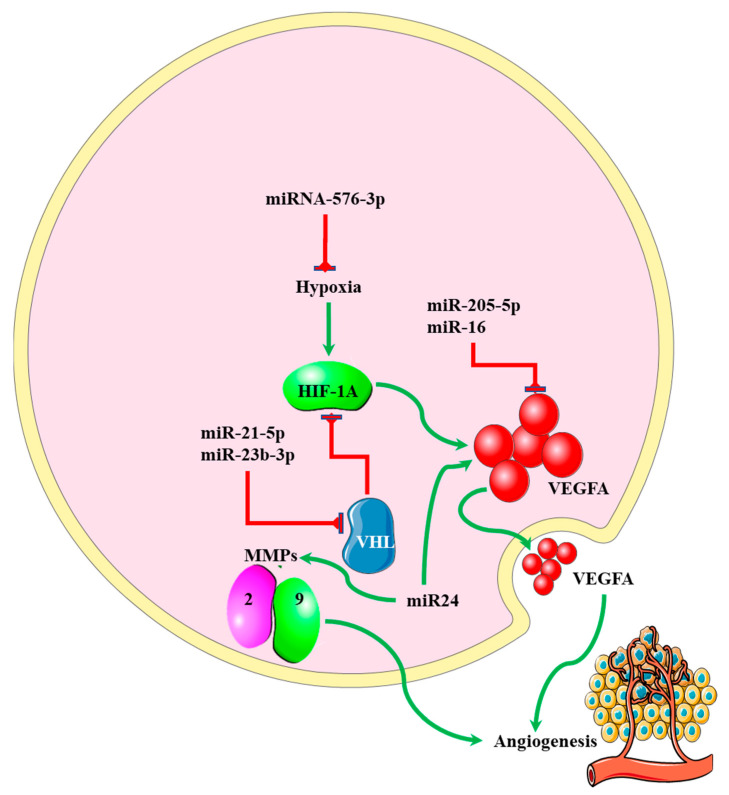
The role played by miRNAs in regulating angiogenesis in glioblastoma. Upregulated miR-24 induced the expression of VEGF, TGF-β, and MMP-2 and -9, leading to increased angiogenesis. VEGF expression is also downregulated by miR-16. Angiogenesis is stimulated by hypoxia, and this process is inhibited by miR-576-3p.

**Table 1 genes-13-00244-t001:** AngiomiRNAs involved in both angiogenesis and EMT in glioblastoma.

miRNA	Effect on Angiogenesis	Effect on EMT
miR-21	Enhanced [100,101]	Enhanced [102]
miR-139-5p	Suppressed [103]	Suppressed [68]
miR-378	Enhanced [104]	Suppressed [67]

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
