# Peer review of "MicroRNA Interrelated Epithelial Mesenchymal Transition (EMT) in Glioblastoma"

_genes, 2022, doi:10.3390/genes13020244_

Round 1

Reviewer 1 Report

Nice paper. I liked it. I would recommend it’s publication

Reviewer 2 Report

Setlai et al. in this review article discussed the role of miRNAs in EMT. They did a good job in providing apt background on EMT, followed by explaining the role of miRNAs in EMT. This review article covered the significant works in the field and also clearly pointed out the potential therapeutic implications.

Therefore, I recommend the work for publication.

Reviewer 3 Report

The manuscript reviewed critically the current literature about microRNAs and the interrelation with the epithelial-mesenchymal-transition, especially in glioblastoma, one of the most difficult tumors to treat. The authors analyzed miRNAs, their targets, and cargos involved in EMT signaling pathway and their potential role as therapeutic targets for glioblastomas.

General Comments

Please be consistent how to spell miRNAs (instead of MiRNAs) throughout the manuscript and both the figures and the figure legends.

Please be consistent with the use of abbreviations throughout the text by defining each abbreviation when it is first used, then continuing to use the abbreviation or acronym throughout the text. 

Please ensure the consistent use of the Oxford comma, which is the final comma in a list of items. Although it is the author’s choice whether to use the Oxford comma, its use (or disuse) should be consistent throughout the text (e.g., lines 70, 83, 98, 106, 315). 

Please provide references at the following lines: 67, 128.

  1. Introduction: The topic has been presented clearly and concisely. Please provide the “epithelial-mesenchymal-transition” abbreviation (Line 49).

  1. Epithelial mesenchymal transition (EMT) in glioblastoma: this paragraph has been described clearly.

However, few edits are required for a better flow of the manuscript. Please remove abbreviation from paragraph title and consider specifying in the main text of the paragraph.

Line 62: there is no reason to specify “epithelial-mesenchymal-transition (EMT),” here. Please refer to previous comment. 

Line 67: Please double check the verbal tense and conjugate the verb accordingly to the subject.

Line 162 Please double check the punctuation.

  1. “Drivers of EMT in glioblastoma” provides a clear and precise overview of different factors involved in the progression of Epithelial mesenchymal transition in glioblastoma, such as extracellular vesicles, TGF-B pathway, and autophagy.

Line 115: Please double check the verbal tense and conjugate the verb accordingly to the subject.

  1. “miRNAs”: it is not clear whether this section should be listed as part of the previous paragraph or needs to be separated. If this is the case, please list it as n. 4.

The authors provide a good description of several studies of miRNA expression signatures that could be use as independent predictors of clinical outcome in glioblastoma. However, this paragraph would benefit of a better consistency in the miRNAs nomenclature.

Please be consistent with the correct spelling for miR-number throughout the manuscript (e.g., lines 214-228, figure 3, figure 4). Conventionally, abbreviations for miRNAs require hyphen after “R” before the number, and eventually and hyphen between the “number” and 5p or 3p. Please modify accordingly.

  1. MiRNA-EMT related angiogenesis. Authors provided a good description of the angiogenesis related to the tumor with a clear figure describing the miRNAs involved in the process.

  1. Conclusions: n/r.

Line 383: First letter of the statement is bold. Please format the sentence properly.

Figures

  • Figure 3: Please consider move A on the top left of the panel.

  • Figure 4: needs some edits.

Please move miRNAs form the cell membrane to the intracellular spaace to increase the readability oif the figure.  Please be consistent with the correct spelling for miR-number.

The green square should be better centered around MMPs. Some arrows and inhibition lines touch the targets, and some don’t. They all should be equally spaced among the figure.

  • Figure 5: Some arrows and inhibition lines touch the targets, and some don’t. They all should be equally spaced among the figure. Please be consistent with the correct spelling for miR-number.
